# Antimicrobial Activity of Gelatin Nanofibers Enriched by Essential Oils against *Cutibacterium acnes* and *Staphylococcus epidermidis*

**DOI:** 10.3390/nano13050844

**Published:** 2023-02-24

**Authors:** Renata Uhlířová, Denisa Langová, Agáta Bendová, Michal Gross, Petra Skoumalová, Ivana Márová

**Affiliations:** Institute of Food Science and Biotechnology, Faculty of Chemistry, Brno University of Technology, 61200 Brno, Czech Republic

**Keywords:** nanofibers, antimicrobial activity, acne vulgaris, essential oils, *Lavandula angustifolia*, *Mentha piperita*, local treatment

## Abstract

Acne vulgaris is a prevalent skin condition that is caused by an imbalance in skin microbiomes mainly by the overgrowth of strains such as *Cutibacterium acnes* and *Staphylococcus epidermidis* which affect both teenagers and adults. Drug resistance, dosing, mood alteration, and other issues hinder traditional therapy. This study aimed to create a novel dissolvable nanofiber patch containing essential oils (EOs) from *Lavandula angustifolia* and *Mentha piperita* for acne vulgaris treatment. The EOs were characterized based on antioxidant activity and chemical composition using HPLC and GC/MS analysis. The antimicrobial activity against *C. acnes* and *S. epidermidis* was observed by the determination of the minimum inhibitory concentration (MIC) and minimum bactericidal concentration (MBC). The MICs were in the range of 5.7–9.4 μL/mL, and MBCs 9.4–25.0 μL/mL. The EOs were integrated into gelatin nanofibers by electrospinning and SEM images of the fibers were taken. Only the addition of 20% of pure essential oil led to minor diameter and morphology alteration. The agar diffusion tests were performed. Pure and diluted Eos in almond oil exhibited a strong antibacterial effect on *C. acnes* and *S. epidermidis*. After incorporation into nanofibers, we were able to focus the antimicrobial effect only on the spot of application with no effect on the surrounding microorganisms. Lastly, for cytotoxicity evaluation, and MTT assay was performed with promising results that samples in the tested range had a low impact on HaCaT cell line viability. In conclusion, our gelatin nanofibers containing EOs are suitable for further investigation as prospective antimicrobial patches for acne vulgaris local treatment.

## 1. Introduction

The skin is the body’s largest organ (approx. 1.8 m^2^). It possesses numerous functions from barrier function, heat regulation, protection of organs from the harmful surrounding environment, etc. The skin is naturally inhabited by numerous microorganisms referred to as skin microbiome [1,2]. The estimated density of these microbial communities is 1 million bacteria/cm^2^. The representation of individual species varies from the individual body part. At the genus level, *Curibacterium* (*Propionibacterium*) and *Staphylococcus* are commonly identified together with *Corynebacterium* [1].

Disbalance of the skin microbiome can easily lead to skin diseases one of the most common is acne vulgaris [3], affecting over 80% of adolescents [4] and approximately 50% of adults older than 25 years. In this older acne group, we can distinguish two separate populations of patients. The first one developed acne during adolescence and have persistent acne and the second group developed de novo acne during adulthood [5]. Acne is manifested by comedones, papules, pustules, and cysts. In the onset of acne, not only one, but several factors apply to its development [6].

In recent years, not only *C. acnes* but other bacteria are studied concerning acne vulgaris. The microbiology of the pilosebaceous unit involves three coexisting groups of microorganisms: gram-positive, coagulase-negative cocci (staphylococci and micrococci); anaerobic diphtheroids (*Cutibacterium acnes* and *Cutibacterium granulosum*), and lipophilic yeasts (*Pityrosporumspecies*). The microflora of comedones is qualitatively identical to that of the normal sebaceous follicle [6]. Other commonly found species in pilosebaceous units included *Staphylococcus epidermidis*, *Cutibacterium humerusii*, and *Cutibacterium granulosum*, each representing 1–2.3% of the total clones [7].

Acne pathogenesis is a multi-factor disease, as both skin microbiota and inflammation play a role in it. The pathophysiology of acne involves abnormal follicular hyperkeratinization, excessive sebum production, colonization by *C. acnes* [8], and subsequent activation of an inflammatory cascade [9]. Moreover, acne demonstration can induce lower self-attitude and self-worth not only in teenagers but adults as well [10].

The genus *Cutibacterium*, formerly *Propionibacterium* [11], constitutes aero-tolerant anaerobic gram-negative bacilli designated for the cutaneous species [11]. *Cutibacterium acnes* constitute an important part of the normal flora of human skin, living in and around sweat glands and sebaceous glands [12]. Under physiological conditions, *C. acnes* is normally a commensal bacterium on the surface of healthy skin. They anaerobically proliferate deeply within follicles and pores by metabolizing sebum triglycerides from the surrounding skin tissues and using them as a primary source of energy and nutrients [11]. The second very abundant bacteria displayed on the skin surface is gram-positive *Staphylococcus epidermidis*. They are facultative anaerobic microorganisms usually involved in superficial infections within the sebaceous unit [6].

Today, there are many ways how to treat acne vulgaris. Topical antibiotics and or chemical peeling agents are being used currently in acne vulgaris treatment; however, oral antibiotics, retinoids, or hormones are prescribed daily [13]. Acne pharmacotherapies are effective but are associated with adverse events such as mood disorders, antibiotic resistance and can leave the skin dry with irritation feelings [14]. Benzoyl peroxide and salicylic acid are non-antibiotic pharmaceuticals frequently administrated in a way of creams to decrease the risk of developing antibiotic resistance and to reduce inflammation [15].

The first description of antimicrobial resistance to antibiotics for *C. acnes* was in 1979 [16] Due to the development of resistant bacterial strains, the number of publications on the antimicrobial activity of phytochemicals is increasing [13,17]. One very promising plant-derived materials are essential oils.

Essential oils (EOs) are volatile, natural, complex compounds characterized by a strong odor and are formed by aromatic plants as secondary metabolites. They are usually obtained by steam or hydro-distillation first developed in the Middle Ages by Arabs [18]. As natural products, they possess interesting physicochemical characteristics with high added values respecting the environment. EOs also have diverse and relevant biological activities such as bactericidal, virucidal, and fungicidal, which have gained them a renewed interest in many areas [19]. Initially, they were discovered and utilized in traditional medicine and used as fragrances and food preservatives not only because of their antibacterial capabilities [20].

EOs as natural products suffer from different quality, quantity, and composition mainly due to cultivation conditions such as climate, soil composition, plant organ, age, and vegetative cycle stage [21]. To ensure the product quality and composition, the whole process must be held under conservative conditions (organ of the plant, soil, climate, and harvest season). Most of the commercialized essential oils are analyzed by gas chromatography and mass spectrometry analysis. Analytical monographs have been published (European Pharmacopeia, ISO, WHO, Council of Europe) [22].

A decrease in activity during storage can be a major drawback of many naturally active compounds. Applications of EOs in the encapsulated form to prolong their stability and to maintain biological activity are seen more often, for example in form of nanofibers for food packaging [23], in nanoemulsions, chitosan nanoparticles [20], liposomes and solid lipid nanoparticles [19] and many others.

Encapsulation technology with nanometric sizes has been remarkably developed to increase the stabilization, activity, and target delivery of bioactive components [24]. Among various nanostructure carriers, such as nanoparticles, nano-platelets, nanoplatelets, nanorods, nanowires and nanofibers, the last mentioned have the ability to create films and be used as patches, protection packaging, filters, and many others. Due to the high surface-to-volume ratio, nanofibers have unique properties such as superior mechanical strength and high porosity with different pore sizes [23,24,25,26].

Various techniques have been developed to obtain nanofibers. Among these techniques belong wet spinning, dry spinning, met pinning, self-assembly, and electrospinning and force spinning [27].

Electrospinning, as one of the simplest methods, holds many other advantages as well. Some of them are the ability to control fiber properties (e.g., diameter, porosity, and morphology) and possible scale-up [28]. The typical setup comprises a polymer that is transferred via a pump through a nozzle tip. By applying a high voltage between the nozzle and the collector, an electrical field is formed which ends in the formation of a Taylor cone, and fibers are collected over the surface of the collector [29]. Recently, some innovative electrospinning techniques were introduced, such as a simple, quick, and cost-effective strategy to manufacture nanofiber-constructed yarns combining conventional electrospinning technique with hand winding and stretching [30] or preparation of tryptophan-based poly(ester urea)s composite mats with good biodegradation and biocompatibility [31]. Another intriguing technique is solution blow spinning, in which the electrical field is replaced by the high-velocity gas flow. Since a high-voltage power source is not required, the collector could be nearly anything, from common flat collectors to living tissue such as a human hand [32].

There has been increased interest in the production of natural-based materials including nanofibers due to their bio-comparability and non-toxicity. One of the integrating food grade spin-able biopolymers is gelatin. It is the most abundant structural protein found in animal connective tissues such as skin, tendon, cartilage, and bone. Gelatin has multiple functions such as enhancing food products’ elasticity, stability, and consistency [33].

In this study, we aimed to the determination of the antimicrobial activity of two essential oils: *Lavandula angustifolia* and *Mentha piperita.* This was achieved by determination of the minimal inhibition and bactericidal concentrations (MIC and MBC) by the broth microdilution method, metabolic resazurin assay, and agar pinch method against *C. acnes* and *S. epidermidis.* Characterization of EOs was carried out by determining antioxidant activity using TEAC assay, HPLC analysis of allergens, and GC/MS analysis of chemical composition. Subsequently, essential oils were encapsulated into gelatin nanofibers to ensure proper dosage, and their antimicrobial properties were tested by agar diffusion tests as well on both bacterial strains. Moreover, a disc diffusion test of pure EOs and 20% EOs in almond oil was carried out to compare with data from agar plate tests with gelatin nanofibers. Finally, a scanning electron microscope (SEM) was utilized to photograph the nanofibers that had been created, as well as to analyze and compare fiber diameter and morphology.

## 2. Materials and Methods

### 2.1. Chemical Composition of Essential Oils

Commercial essential oils from *Mentha piperita* and *Lavandula angustifolia* were purchased from Saloos naturcosmetic (Blansko, The Czech Republic) marked in the following text and tables with the letter “S” (plant origin: peppermint India and lavender France—Provence) and Fichema Inc. (Brno, The Czech Republic) marked in the following text and tables with the letter “F” (plant origin: Peppermint from Spain, lavender from Bulgaria). Gelatin was purchased from Penta, Czech Republic. The composition, according to the manufacturers, is listed in Table 1 and Table 2.

For characterization of the chemical composition of essential oils from *Mentha piperita* and *Lavandula angustifolia* besides composition obtained by the manufacturer, antioxidant activity TEAC assay, HPLC analysis of allergens, and composition by GC/MS have been performed.

#### 2.1.1. Antioxidant Activity

For evaluation of antioxidant activity, Trolox equivalent assay with ABTS^·+^ cation radical was performed [34]. ABTS radical was prepared by mixing 7 mM ABTS (2,2’–azino–bis(3–ethylbenzothiazoline–6–sulfuric acid) diammonium salt) (Sigma–Aldrich, St. Louis, MO, USA), with 2.45 mM potassium persulphate (Sigma–Aldrich, St. Louis, MO, USA) in deionized water and allowed the mixture to stand in the dark at room temperature for 12–16 h before use. Before the start of the analysis, ABTS^·+^ solution was diluted with UV-vis ethanol for absorption A = 0.70 ± 0.02 at 734 nm with ethanol as blank. For measurement of antioxidant activity, 1 mL of diluted ABTS^·+^ and 10 μL aliquot of sample was added into the Eppendorf test tube, mixed, and kept in the dark. The decrease in absorption was measured after 10 min. The calibration curve was prepared by Trolox dilution in the range of concentration 50–400 μg/mL. Values are represented as mean ± standard deviation (SD) of three replicate determinations.

#### 2.1.2. Determination of Allergens Content by HPLC

As allergens are common compounds that can be found in essential oils, HPLC analysis with the reversed phase was performed according to [35]. Analysis was performed on HPLC/PDA Dionex UltiMate 3000 (Thermo Fischer Scientific, Waltham, MA, USA), DAD detector Vanquish series (Thermo Fischer Scientific, Waltham, MA, USA), column YMC-Triart C18 ExRS S-3 µm, 8 nm Analytical HPLC Column, 150 mm × 4.6 mm.

#### 2.1.3. Composition Determined by GC/MS

HS-SPME-GC-MS was employed for analysis of major components on Trace^TM^ 1310 split/splitless injector (Thermo Fisher Scientific Inc., Waltham, MA, USA) with mass detector ISQTM LT Single Quadrupole (Thermo Fisher Scientific Inc., Waltham, MA, USA). Sample extraction conditions were as follows: 10 min incubation at 40 °C, 20 min extraction at 40 °C to PDMS/CAR/PVB fiber injector temperature 240 °C, and desorption time 20 min. Analysis was performed on the capillary column TG-WaxMS, 30 m × 0.25 mm × 0.5 μm. Helium was used as carrying gas at a flow rate of 1 mL·min^−1^. The temperature program was set for 40 °C for 1 min and 5 °C/min until 220 °C with 12 min hold. The mass detector was in electron ionization mode and set for 70 eV, ion source temperature 200 °C, scanning range 30–400 m/z, scanning speed 0.2 s.

### 2.2. Antimicrobial Susceptibility Tests

Microorganisms were obtained from the Czech Collection of Microorganisms (CCM, Czech Republic) namely *Cutibacterium acnes* CCM 3437 and *Staphylococcus epidermidis* CCM 4418. Lyophilized cultures were rehydrated and cultured according to manufacturer recommendations. Both microorganisms were cultivated on BHI media (Himedia, Brno, Czech Republic) at 37 °C in (KS 4000 I control, IKA, Königswinter, Germany) for 24 h. Microorganisms were subsequently seeded into fresh media at the concentration of 0.5 McFarland units (approximately 1.5·10^8^ cells/mL). In the case of cultivation on solid media, 20 g/L of microbiological agar (Sigma-Aldrich, St. Louis, MO, USA) was added, before the sterilization, to the media.

For the purpose of this study, preliminary broth dilution tests on a 96-well plate were carried out to determine whether selected essential oils possess good antimicrobial activity and are suitable for encapsulation into electrospinning nanofibers. The Minimal Inhibition Concentration (MIC) and Minimal Bactericidal Concentration (MBC) were determined for all EOs samples. Moreover, disc diffusion tests of pure and 20% diluted EOs in almond oil were performed. In the case of nanofibers with essential oils, diffusion tests on agar plates were performed as well. The goal was to assess whether gelatin nanofibers can be used as carriers of antimicrobial compounds of proper EOs dosages, which will not affect surrounded micro-biome and skin and to only act on the spot of application.

#### 2.2.1. Preliminary Broth Microdilution Tests Minimal Inhibition Concentration (MIC) Assay

For the first screening, the broth dilution method was chosen and performed on a 96-well plate. Both *Cutibacterium acnes* (CA) and *Staphylococcus epidermidis* (SE) cultures were firstly cultivated in Erlenmeyer flasks and the 24 h culture was then sub-cultured by adding 100 μL of diluted cultures onto a 96-well plate. Before the analysis, each sample was diluted 5 times in ethanol for UV-vis spectrophotometry. Subsequently, a 2-fold dilution was prepared in the cultivation media to obtain the concentration range of 0.012–25 μL/mL. As positive control that possesses antimicrobial activity, ampicillin (Sigma-Aldrich, St. Louis, MO, USA) was used at starting concentration of 25 μg/mL. Prepared 96-well plates were read on a well plate reader (Synergy HTX, Agilent Technologies, Inc., Santa Clara, CA, USA) at 630 nm and placed into an incubator, and cultivated for 24 h at 37 °C. In the case of *C. acnes*, all wells were sealed to ensure anaerobic conditions. Afterward, the absorption was measured again at 630 nm. Antimicrobial activity was calculated as a percentage of control with non-treated microbial culture. Each sample dilution was tested in triplicates. The antimicrobial efficiency is described as MIC_90_ concentration at which 90% of the microbial cells were inhibited [36].

#### 2.2.2. Resazurin Reduction Assay

The metabolic assay was performed using a resazurin reduction assay. After 24 cultivation of samples with microbial culture, 20 μL of resazurin solution (0.15 mg/mL in PBS) was added into each well. Microplates were placed back into the incubator for 30 min at 37 °C. After incubation, the plates were read at 570 nm as the conversion of resazurin to resorufin is associated with a shift in absorbance spectrum (resazurin: λ_max_ = 600 nm, resorufin: λ_max_ = 570 nm) [37]. Absorbance from control wells with media only was deducted from each well. Metabolic activity was calculated as a percentage of positive control (media with MO). Each sample dilution was tested in triplicates. A change from blue to pink indicates the reduction of resazurin and therefore bacterial growth. The MIC was defined as the lowest drug concentration that prevented this color change [38].

#### 2.2.3. Minimal Bactericidal Concentration Test (MBC)

For determination of minimal bactericidal concentration samples after 24 h cultivations were transferred from 96-well plates onto BHI agar plates using a sterile inoculation needle and incubated for another 24 h at 37 °C [17]. After cultivation, bacterial colonies were observed and visualized by a camera. MBC is defined as the lowest concentration that completely inhibited bacterial growth [39]. Bactericidal concentration was evaluated as the lowest concentration at which no bacterial colony was observed after 24 h of cultivation. Each determination was performed in triplicate.

#### 2.2.4. Agar Diffusion Tests

In the case of *Staphylococcus epidermidis*, 100 μL of 24 h culture was inoculated and spread on the surface of the agar plate by a cell spreader. This type of inoculation was used to assess whether the antimicrobial effect of samples is only local (on the spot of application) and does not alter the viability of surrounding bacterial cells. Essential oils were encapsulated into gelatin nanofibers and tested for antimicrobial activity by agar diffusion test [40]. After 24 h incubation at 37 °C, inhibition zones were observed and measured.

In the case of *Cutibacterium acnes*, the 24 h culture was inoculated with an inoculation needle on agar in several punctures to mimic the conditions where *C. acnes* proliferates anaerobically deep in the follicles and pores by metabolizing sebum triglycerides from the surrounding skin tissues. This type of experiment was carried out to confirm that active compounds from samples are released into the agar and act in the depth of the inflamed follicle. Gelatin nanofibers with encapsulated EOs were placed on one of the punctures to observe the local antimicrobial effect against *C. acnes*. After 24 h incubation at 37 °C, the growth of the bacteria in covered punctures and the possible affection of the uncovered punctures were observed.

Besides these samples, pure essential oils and 20% essential oils were diluted in almond oil (20% *v*/*v*) and 10 μL were applied on the diffusion disc and placed on a BHI agar plate with inoculated bacterial cultures. Ampicillin stock solution (25 μg/mL) was used as a positive control that possesses antimicrobial activity. 100% almond oil was used as a negative control that had no antimicrobial activity on selected bacterial strains. All samples were made in triplicates.

### 2.3. Preparation of Gelatin Nanofibers

For nanofiber preparation, the electrospinning technique was employed. Nanofiber films loaded with essential oils were prepared as follows: gelatin solution 30% (*w*/*v*) was prepared in 50% acetic acid [41]. Nanofibers loaded with essential oils were prepared by additions of 20% (*w*/*w*) of nanofiber material, stirred for 10 min, and placed into a syringe with a metallic needle with a diameter of 1 mm.

Nanofibers were prepared on the in-house assembled device that consisted of a high-voltage source, NE-1010 Higher Pressure Syringe Pump (KF Technology SRL Italy), and the metal collector (25 cm × 25 cm) covered with aluminum foil.

Each sample was collected on the new aluminum sheet. The positive charge was placed on the tip of the needle through the spinning solution, pushed by a syringe pump. The negative charge was placed on a collector. The electrospinning condition was set as follows: 15 cm distance of the needle from the metal collector, 15 kV, needle diameter of 1 mm, and temperature of 22 °C with humidity of approximately 40–50%. After collection, all samples were weighed and stored in plastic sample bags at laboratory temperature.

### 2.4. Nanofiber Characterization

In order to characterize prepared nanofibers from gelatin, scanning electron microscopy was employed. Microscope ZEISS EVO LS 10 (ZEISS, Jena, Germany) was operated under high-vacuum mode and acceleration voltages of 5 kV. The sample surfaces were sputtered for 1 min with Au. Fiber diameter was measured as well by the microscope software SmartSEM (ZEISS, Jena, Germany) tool.

### 2.5. Cytotoxicity Assays

For MTT cytotoxicity assays human epidermal keratinocyte cell line (HaCaT) was obtained from CLS Cell Lines Service GmbH (Eppelheim, Germany). Cells were maintained in Dulbecco’s Modified Eagle Medium (DMEM) (Lonza Biotec, Kouřim, The Czech Republic) with High Glucose, with 0.4 mM L-Glutamine, without Sodium Pyruvate, supplemented with 100 mg/mL of streptomycin, 100 units/mL of penicillin, and 10% of heat-inactivated fetal bovine serum (FBS). Cultivation was performed in humidified, 5% (*v*/*v*) CO_2_ atmosphere at 37 °C. Cell culture was fed every 2–3 days and passaged after reaching 80% of confluence [42].

### 2.6. MTT Assay

Tested samples were treated according to Li et al. as follows: samples were diluted, pipetted through a 200 nm syringe filter, and diluted with DMEM until the concentration range of 0.012–0.75 μL/mL.

Firstly, 100 μL of properly diluted cell culture was added to a 96–well plate and left in the incubator. After 24 h medium was replaced by prepared samples in DMEM and placed back into the incubator. The control was DMEM medium, and 60% ethanol served as a negative control. Subsequently, after another 24 h 20 μL of MTT dissolved in PBS (2.5 mg·mL^−1^) was added to each sample and incubated for 3 h in an incubator, and then 100 μL of 10% SDS in PBS was added to each well. Plates were stored in dark and evaluated the next day by ELISA Reader at 543 nm [43].

### 2.7. Statistical Analysis

All experiments were performed in triplicates. Results are presented as mean ± standard deviation (SD). For hypothesis testing, a one-way ANOVA was used and set at a significance level of 0.05. Data were evaluated by Tukey’s HSD test and marked as groups that had significant differences at levels 0.05; 0.001 or 0.000 1.

## 3. Results and Discussion

### 3.1. Chemical Characterization of Essential Oils

Several approaches were used for chemical characterization, including the TEAC assay for antioxidant activity, HPLC study of allergens, and GC/MS profile of compounds. The chemical composition reported by manufacturers was compared to the results of our analyses.

#### 3.1.1. Antioxidant Activity

Antioxidant activity is together with antimicrobial activity one of the most studied biological activities for essential oils [40,44,45]. The antioxidant activity of *Mentha piperita* and *Lavandula angustifolia* is listed in Table 3.

In the case of lavender samples, there was no statistical difference between their antioxidant activity. Both samples showed moderate values around 0.035 mmol TE and were in the range of 15.37 to 16.37 mg/mL for 50% scavenging concentration. Other researchers, such as Badr et al. [46], determined IC50 for lavender EO as 1 828.25 mg/L so 1.83 mg/mL. The other study of eight different lavender samples showed IC50 in the range of 61.23 to 74.08 μg/mL in ABTS scavenging assay [47]. The study of Al-Ansari et al. showed that IC50 for *L. latifolia* was around the value 0.3 mg/mL with ABTS^+·^ and 0.35 mg/mL of essential oil for DPPH radical [48] and *L. dentata* IC_50_ of 14.03 ± 0.16 mg/mL in the study of Demmak et al. [49]. In the case of *Lavandula* genus, there was reported a similar or bigger variance in antioxidant activity by other researchers [50,51] than in the case of our samples.

In the case of mint samples, there was a significant difference between their antioxidant activity. The *p*-Value for these two samples was *p* < 0.000 1. Fichema sample originated from Spain and had SC50 of value 1.90 ± 0.02 mg/mL, while Saloos sample originated from India and exhibited 3.64 ± 0.27 mg/mL. As this value is almost double, the sample from India had half the antioxidant activity. After comparing the literature, we realized that our findings are in accordance with the study of Mollaei et al. 2020, where they compared twelve wild populations of *Mentha puleguim* from *Azerbaijan.* Their results showed that antioxidants activity ranged for IC50 between values 0.545–4.884 mg/mL [52].

#### 3.1.2. Quantitative and Qualitative Analysis of Essential Oils

The chemical components of essential oils were defined through the analysis of allergens by HPLC and the chemical composition of GC/MS. For allergens analysis, HPLC with reversed phase was performed and the data are shown in Table 4. From 24 tested allergens, all samples contained only two of them, linalool and limonene. Lavender samples contained a high amount of linalool–441.41 mg/mL (Fichema sample from Bulgaria) and 479.05 mg/mL (Saloos sample from Provence). These results are supported by data obtained from manufacturers shown in Table 1 and Table 2 as linalool is a major component of lavender essential oil [40,53,54,55]. From other allergens, only limonene was declared by the manufacturer as well, but only in the case of lavender from Bulgaria. In this sample, 21.14 mg/mL of limonene was detected. In this sample, citral was detected as well, but it was not listed in the chemical composition of the manufacturer.

Allergen analysis for mint samples showed that the content of allergens was not consistent. However, in the case of eugenol, both samples contained a small but similar amount of this allergen, whereas, in both lavender samples, eugenol was not detected. In other studies, eugenol was detected by GC/MS at a level of 0.02–0.3% [56,57].

Besides the data obtained from HPLC analysis of allergens, GC/MS data can provide a deeper understanding of the chemical composition of essential oils. For a comparison of the chemical composition declared by the manufacturer, we decided to perform GC/MS analysis. Measured data from GC/MS qualitative analysis are displayed in Appendix A. We were able to identify 50–66 components in each sample, with only 17 of them being present in all of them. There are 65 components in Bulgarian lavender samples and 59 components in Provence lavender samples. In both lavender samples, there was a 56-component overlap. Mint samples from India contained 50 components, while mint samples from Spain contained 62 components, and both mint samples contained 40 components.

Table 1 and Table 2 demonstrate the chemical composition of lavender and mint samples as declared by manufacturers. Table 4 shows the main components presented in EOs as detected by GC/MS analysis. Eucalyptol was the main component present in all tested samples. Other components were present only in two samples, but not necessarily from the same plant. Components such as linalyl butyrate and b-caryophyllene were determined in lavender from Provence, but the same compounds were present both in mint from India and Spain.

### 3.2. Antimicrobial Activity Testing

The presence of volatile chemicals such as terpenes, alcohols, acids, esters, epoxides, aldehydes, ketones, amines, and sulfides terpineol, thujanol, myrcenol, neral, thujone, camphor, and carvone contribute to the essential oil’s biological activity [58]. Both broth microdilution and the agar diffusion test were used to determine antibacterial activity.

#### 3.2.1. Preliminary Broth Dilution Tests—Minimal Inhibition Concentration (MIC) Assay

The antibacterial properties of *Lavandula angustifolia* and *Mentha piperita* EOs samples were tested against *C. acnes* and *S.epidermidis* and evaluated by determining minimum inhibitory and bactericidal concentrations (MICs and MBCs, respectively). Data are shown in Table 5.

For lavender oil, Martucci et al. 2015 determined MIC for *E. coli* strain value of 2 μg/mL [54]. In the study of multiple essential oils’ antimicrobial activity against *Staph. aureus* strain EG-AE1, *Staph. epidermidis* strain EG-AE2, and *Cutibacterium acnes* Strain EG-AE1, they came to the conclusion that lavender essential oil does not have any antimicrobial activity in disc diffusion tests [17]. This contrasted with our findings, where MIC for both strains and both lavender samples were in the range of 6.3 to 9.4 μL/mL. As expected, minimal bactericidal concentration was higher for both lavender samples and bacterial strains and ranges from 12.5 to 25 μL/mL.

In comparison to lavender samples, both mint samples had equivalent or superior antibacterial activity. This phenomenon could be explained by the presence of a minor amount of eugenol in both mint samples, according to the chemical composition data (Table 2). Eugenol was the main ingredient in many types of essential oils produced from plants, and it had remarkable antibacterial action that had been well-studied and employed in food preservation, and the US Food and Drug Administration (FDA) has classified it as generally recognized as safe (GRAS) [33]. Another antimicrobial compound derived from *Mentha piperita* mint oil is carvone. This compound destabilizes the permeability of cell membranes, diminishes the ergosterol rates, and consequent reduction throughout the production of PM-ATPase [45]. MIC values against *E. coli* were in the range of 6.25–12.5 mg/mL and the same values were measured against *S. aureus* for differently treated *Mentha* species [59].

Based on data obtained for each bacterial strain, we can say that values of MIC and MBC are lower in the case of *S. epidermidis* in comparison to *E. coli*. This is a common phenomenon that is based on cell wall composition. In the case of gram-negative bacteria, the lipopolysaccharides outer layer permits small hydrophilic molecules and is partially permissive for hydrophobic molecules. Contrarily, Gram-positive strains such as *S. epidermidis* are more sensitive to essential oils based on the structure of their cell wall. Peptidoglycans in the cell wall allow the hydrophobic molecules to enter the cell [60,61]. Antibacterial agents can act in different ways. Leakage of specific cell markers, such as proteins, can indicate cytomembrane damage and integrity of cells treated by antimicrobial agents [62]. The effect of essential oils is caused predominantly by simultaneous membrane disruption that causes leakage of intracellular substances including proteins and K^+^ [63].

#### 3.2.2. Discs Diffusion Test of Pure and Diluted EOs

Essential oils in pure form and diluted in almond oil at 20% (*v*/*v*) were tested against *S. epidermidis* and *C. acnes*. Inhibition zones are displayed in Table 6 and represented as the mean of three independent samples with standard deviation.

When compared to the control with almond oil, both lavender and mint samples exhibited visible inhibitory zones. There was no significant difference between the two lavender samples as well as between mint samples in pure and diluted form.

The evaluation of the diffusion test is different in the case of *Staphylococcus epidermidis* and *Cutibacterium acnes*. As mentioned above, *Staphylococcus epidermidis* is an opportunistic pathogen that is often found on the surface of the skin. *Cutibacterium acnes* is one of the most common bacteria responsible for acne manifestation and its multiplication occurs mainly in the anaerobic environment of the follicle. The purpose of this test was to find out whether the prepared nanofibrous disc is suitable for the local elimination of *S. epidermidis* growing mainly on the skin surface. Further, we would like to verify the release of the active substance in depth to eliminate *C. acnes* as well.

In Table 6, we can see that even when we diluted essential oils 5 times to 20% in almond oil, the inhibition zone diameter did not decrease 5 times but less. In the case of both *Lavandula angustifolia*, the reduction of inhibition zones was not 80% as expected, but only 58.9 ± 2.4%. while diluted *Mentha piperita* essential oils exhibited a reduction of inhibition zones by 72.3 ± 3.7%. As all samples were diluted in almond oil, pure almond oil was tested as well, and no antimicrobial zone was observed.

Antimicrobial testing against *C. acnes* revealed that all samples in both dilutions exhibited inhibition zones. While this testing was performed on agar plates with needle inoculation, the inhibition zones diameter was evaluated only semi-quantitatively and was displayed as the ++ sign if the inhibition zone was observed and affected both central and surrounding colonies, the + sign if the sample affected only the central colony, or the —sign when there was no visible inhibition zone (Table 6).

As the intended application of tested essential oils is as patches that will be placed only over the spot with pimples or lesions, the desired antimicrobial activity should be as well only on the spot of application. Therefore, the nanofibers patch should act locally with minimal effect on surrounding skin microbiota. For pure essential oils, both lavender samples had a local antimicrobial effect against *C. acnes*, while pure mint essential oils affected even surrounded colonies of *C. acnes*. A local effect was observed after dilution to 20% in almond oil for all samples.

There are only limited data on the antimicrobial activity of essential oils against *C. acnes*. In the work of Esmael et al. 2020 they tested several plant oils and from tested samples, only tea tree and rosemary oil exhibited antimicrobial activity. For the tea tree, the inhibition zone was 20.85 ± 0.76 mm and 14.77 ± 0.35 mm for rosemary oil. Very similar data were obtained in this study for antimicrobial activity against *S. epidermidis* [17]. Our data for *S. epidermidis* were very similar to data from this research in the case of *Lavandula angustifolia*, but even higher inhibition zones were observed in our *Mentha piperita* samples.

Blažeković et al. 2018 performed antimicrobial disc diffusion tests of *Lavandula angustifolia* samples on several gram-positive and gram-negative bacteria. For gram-positive bacterial strains, the inhibition zones for *L.* × *intermedia* ‘Budrovka’ were 15–23 ± 2 mm and 14–19 ± 1 mm, while inhibition zones for *L.* × *intermedia* ‘Budrovka’ and *Lavandula angustifolia* against *S. aureus* were 20 mm and 18 mm, respectively. In the case of gram-negative bacterial strains, the data suggest that the inhibition zones were ranging from 14 to 22 ± 2 mm for *L.* × *intermedia* ‘Budrovka’ and 12–17± 1 mm for *Lavandula angustifolia* [39].

Another study by Yuan et al. 2019 showed that the inhibition zones diameter for *Lavandula angustifolia* grown in China in disc diffusion tests was 20.22 ± 0.7 mm in the case of *S. aureus* and 19.86 ± 0.9 mm in the case of *E. coli* [40].

Despite several studies and reviews about the antimicrobial activity of *Mentha* components such as carvone [45] and others [64], there are only limited data on antimicrobial activity and even less against *C. acnes*. Zu et al. 2010 tested ten essential oils for antibacterial activity against *C. acnes* and they found that MIC and MBC concentrations were 0.25% (*v*/*v*) for *Mentha* essential oil [65].

In another study by Desam et al. 2019, inhibition zones were measured as the evidence of antimicrobial activity of *Mentha piperita* essential oil against several bacterial strains. Authors determined inhibition zones of 35.14 ± 0.08 mm against *S. epidermidis* and 27.02 ± 0.13 mm against *E. coli* [57].

#### 3.2.3. Antimicrobial Activity of Gelatin Nanofibers Analyzed by Diffusion Test

After verification that selected EOs possessed antimicrobial activity, the next step was to test the antimicrobial activity of prepared nanofibers with and without EOs. The control agar plate of *C. acnes* after 24 h incubation and the plate with gelatin nanofibers with a 20% addition of lavender F Bulgaria EOs are presented in Table 7. The dots represent where *C. acnes* cells were inoculated by puncture of the agar with an inoculation needle to mimic conditions of the anaerobic proliferation of *C. acnes* deeply within follicles and skin pores. For gelatin nanofiber samples, the area where the sample was placed is marked. As we can see from this picture, gelatin nanofibers without any added EO had no antimicrobial effect on *C. acnes*. The very same we can say for gelatin fibers (Gel0) and their antimicrobial activity against *S. epidermidis*, as is represented in Table 7.

All nanofiber samples tested against *C. acnes* and *S. epidermidis* had desired local effect, whereas ampicillin affected surrounding colonies as well. Therefore, our nanofiber samples with Eos are suitable for the local treatment of acne vulgaris (plates with other tested EOs are shown in Appendix A).

### 3.3. Morphology of Gelatin Nanofibers (Electrospun Meshes)

To observe the morphology of prepared nanofibers, scanning electron microscopy was performed within all samples. As the SEM micrograms in Table 8 are showing, the concentration of essential oils was alternated for three concentrations, namely, 0, 2, and 20% *w*/*w* of the solid material (marked as A–C and 1 for 5000 times magnification and 2 for 2000 times magnification), which had some minor effect on the sample’s morphology. The fibers exhibited a more curved shape when 20 percent essential oil is added, compared to samples with no or 2 percent essential oil, where fibers exhibit higher orientation in the X and Y axis. Fibers were spun in a higher order on the collection surface in these two samples (A and B).

The diameter of prepared nanofibers was measured directly by the microscope software and the data from the five measurements are displayed in Table 9. In all cases, nanofibers were very uniform and had a mean diameter of approximately 430 nm.

In Table 9 we can see the diameters of prepared nanofibers and corresponding *p*-Values. There was a significant difference only after the addition of 20% of essential oil into nanofibers compare to the sample with 2% of essential oil. Altogether with pictures from SEM from Table 8, we can conclude that only a 20% addition of essential oil has an impact on both fiber diameter and morphology of the prepared fibers.

### 3.4. MTT Cytotoxicity Assay

The HaCaT cell line is considered to be a reliable model for the assessment of different skin disorders and is frequently used in the literature to assess the biocompatibility of different EOs [66,67]. In the present study, HaCaT cells were exposed for 24 h to a concentration range of 0.012–0.75 μL/mL of tested essential oils and lethal concentrations LC10, LC50, and LC90 were extrapolated and calculated. Data are expressed as a lethal concentration in μL/mL of added samples and are shown in Table 10. Our results indicated that all samples had only a low impact on HaCaT cell line viability in the tested range of concentration (LC 10 for maximum added concentration 0.70 μL/mL). For the Mint F sample from India, it was not possible to calculate or extrapolate the LC 50 and LC 90 because, for all tested concentrations, the cell viability was around 80–90%. Therefore, the samples in the tested range could be potentially applicable to human skin.

## 4. Conclusions

To summarize, the findings show that incorporating *Lavandula angustifolia* and *Mentha piperita* essential oils into gelatin nanofibers is a suitable method for producing absorbable patches for the local treatment of *Acne vulgaris* without causing any negative effects on the surrounding microbiome. Despite differences in the chemical composition of each sample, antioxidant activity for *Lavandula angustifolia* of different origins was equivalent, while for *Mentha piperita*, the sample from Fichema company (plant origin Spain) had two times the antioxidant activity of the sample from Saloos company (plant origin India). This research backs up previous observations that the country of the plant’s origin caused certain variations in its chemical composition and biological activity. Aside from this, antimicrobial activity data demonstrated that differences in chemical composition between samples had no significant impact on MIC and MBC and that the primary components, which were present in both samples from the same plant, were the main reason. In pure form, all samples demonstrated good antibacterial activity in agar diffusion assays. Although dilution with almond oil reduced antibacterial effectiveness, samples had an impact on bacterial growth in the surrounding area. Only after incorporation into gelatin nanofibers was the dose sufficient for the samples to act as local antibacterial agents against *C. acnes* and *S. epidermidis*, making them suitable for the local treatment of *Acne vulgaris*. MTT assay showed low cytotoxicity of tested samples which together with their antioxidant activity may facilitate their use in the gentle and focused treatment of different skin infections including frequently indicated *Acne vulgaris*.

## Figures and Tables

**Table 1 nanomaterials-13-00844-t001:** Chemical composition of *Lavandula angustifolia* according to the manufacturer.

Compound	Lavender F Bulgaria	Lavender S Provence
Linalool	31.26	36.98
Linalyl Acetate	31.63	42.13
beta-Caryophyllene	3.96	2.37
3-Octanone	1.48	0.54
Bornyl Acetate	0.1	ND
Borneol	ND	1.52
beta-Farnesene	3.96	ND
Lavandulyl Acetate	3.9	ND
Terpinen-4-ol	3.74	ND
Lavandulol	1.77	ND
Ocimene	5.13	ND
Limonene	0.761	ND
Total number of components	65	59

ND—not defined, F—sample from Fichema Inc., S—sample from Saloos, Czech Republic.

**Table 2 nanomaterials-13-00844-t002:** Chemical composition of *Metha piperita* EOs according to manufacturer.

Compound	Mint S India	Mint F Spain
Menthol	48.52	40
Menthone	19.61	10–25
Isomenthone	ND	10–25
Neomenthol/Menthofuran/Isomenthone	8.83	ND
alpha-Pinen	0.27	1–10
beta-Pinen	0.26	1–10
Limonene	1.71	1–10
Pulegone	2.21	5
Methyl acetate	4.4	ND
1,8-Cineol	4.04	ND
beta-Caryophyllene	2.16	ND
Linalool	0.33	ND
Total number of components	50	62

ND—not defined, F—sample from Fichema Inc., S—sample from Saloos, Czech Republic.

**Table 3 nanomaterials-13-00844-t003:** Antioxidant activity of *Mentha piperita* and *Lavandula angustifolia* measured by TEAC method.

Sample	Antioxidant Activity (mmolTE)	SC50 (mg/mL)
Mint F Spain	0.415 ± 0.004 *	1.90 ± 0.02 *
Mint S India	0.217 ± 0.016 *	3.64 ± 0.27 *
Lavender F Bulgaria	0.033 ± 0.026	15.37 ± 0.86
Lavender S Provence	0.037 ± 0.021	16.37 ± 1.12

* Indicates a significant difference between samples of *p* < 0.000 1.; F—sample from Fichema Inc., S—sample from Saloos, Czech Republic.

**Table 4 nanomaterials-13-00844-t004:** Quantitative and qualitative analysis of EOs.

Sample	Lavender F Bulgaria	Lavender S Provence	Mint F Spain	Mint S India
Allergen	(mg/mL)	(mg/mL)	(mg/mL)	(mg/mL)
Coumarin	ND Q	2.36 Q	ND	ND
Linalool	441.41 Q	479.05 Q	17.32 Q	76.29
Citral	18.42 Q	ND Q	6.81	39.04
Limonene	21.14 Q	2.39 Q	20.08 Q	47.64 Q
Geraniol	15.43 Q	5.55 Q	ND	ND
Eugenol	ND	ND	1.89 Q	1.16 Q
Myrcene	Q	ND	ND	ND
Eucalyptole	Q	Q	Q	Q
(Z)-beta-Ocimene	Q	ND	ND	ND
Octenyl Acetate	Q	ND	ND	ND
Linalyl Butyrate	ND	Q	Q	Q
Bornanone	ND	Q	ND	ND
beta-Caryophyllene	ND	Q	Q	ND
Menthone	ND	ND	Q	Q
Menthol	ND	ND	Q	Q

ND—not detected by HPLC quantitative analysis, Q-detected by GC/MS qualitative analysis.

**Table 5 nanomaterials-13-00844-t005:** MIC and MBC of lavender and mint EOs.

	*S. epidermidis*		*C. acnes*	
Sample	MIC (μL/mL)	MBC (μL/mL)	MIC (μL/mL)	MBC (μL/mL)
Lavender F Bulgaria	8.3 ± 2.2	16.7 ± 7.2	9.4 ± 3.6	25.0 ± 2.1
Lavender S Provence	6.3 ± 1.0	12.5 ± 1.0	6.3 ± 1.0	25.0 ± 1.0
Mint F Spain	6.3 ± 3.0	9.4 ± 3.1	6.3 ± 3.4	20.8 ±7.2
Mint S India	7.3 ± 2.1	12.5 ± 1.1	5.7 ± 0.9	14.0 ± 5.6

MIC—Minimal Inhibition Concentration, MBC—Minimal Bactericidal Concentration.

**Table 6 nanomaterials-13-00844-t006:** Antimicrobial activity of essential oils measured by disc diffusion test.

	*S. epidermidis*		*C. acnes*	
Sample	Pure EO	20% in Almond Oil	Pure EO	20% in Almond Oil
	Inhibition Zones (mm)	Inhibition Zones (mm)		
Lavender F Bulgaria	14.0 ± 2.0	6.0 ± 0.5	+,+,+	+,+,+
Lavender S Provence	16.5 ± 0.5	6.5 ± 0.5	+,+,+	+,+,+
Mint F Spain	28.0 ± 1.0	8.5 ± 1.5	++,++,++	+,+,+
Mint S India	26.0 ± 4.0	6.5 ± 0.5	++,++,++	+,+,−

++ visible inhibition zone that affected not only the central colony but the surrounding colony too, + visible inhibition zone with the effect on the central colony and no visible inhibition zone, − no visible inhibition zone.

**Table 7 nanomaterials-13-00844-t007:** Agar diffusion tests of gelatin nanofibers with a 20% addition of EOs.

*C. acnes*—control agar plate	*C. acnes*—lavender F Bulgaria
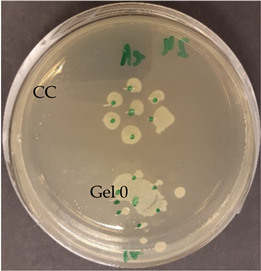	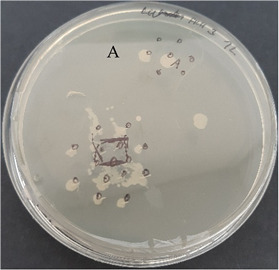
*S. epidermidis*—control agar plate	*S. epidermidis*—lavender F Bulgaria
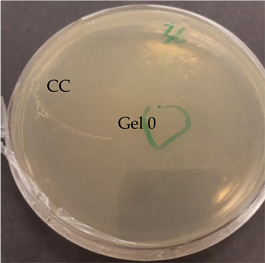	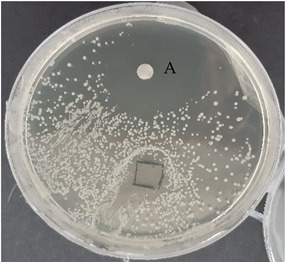

A—ampicillin 25 μg/mL, the marked area represents where nanofibers were placed, CC—culture control, Gel 0—gelatin nanofibers with no added EOs.

**Table 8 nanomaterials-13-00844-t008:** SEM images of the morphology of electrospun gelatin fibers.

A: Gelatin	B: Gelatin with 2% EO	C: Gelatin with 20%EO
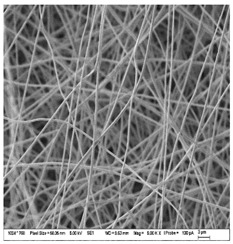	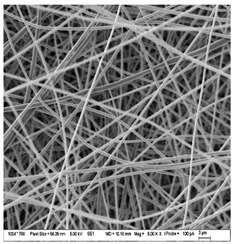	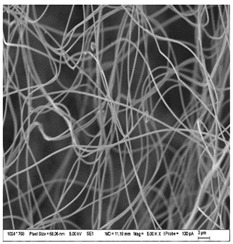
A1: 5000x	B1: 5000x	C1: 5000x
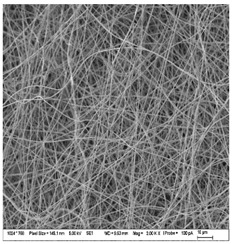	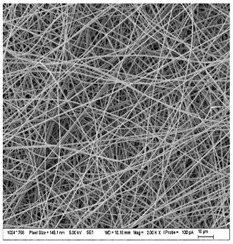	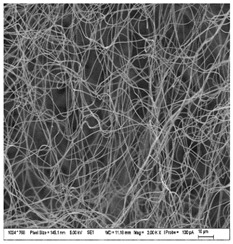
A2: 2000x	B2: 2000x	C2: 2000x

**Table 9 nanomaterials-13-00844-t009:** Diameters of prepared fibers by electrospinning.

Sample	Fiber Diameter (nm)	Treatment Pairs	Tukey HSD *p*-Value	Tukey HSD Inference
Gelatin (A)	430.8 ± 20.6 ^a^	A vs. B	0.082 8	n.s.
Gelatin 2% EO (B)	476.6 ± 51.2 ^b^	A vs. C	0.317 1	n.s.
Gelatin 20% EO (C)	402.6 ± 37.7 ^ab^	B vs. C	0.001 1	** *p* < 0.01

A—stands for gelatin nanofibers, B—stands for gelatin nanofibers with 2% of EO, C—stands for gelatin nanofibers with 20% of EO, and n.s.—stands for not significant results. Samples were sorted into groups by Tukey (HSD) analysis of the differences between the categories A, B, and C, with a confidence interval of 95% marked by letters (a and b). Samples on different significant levels are marked as follows: ** *p* ≤ 0.001 and n.s. for not significant results.

**Table 10 nanomaterials-13-00844-t010:** Cytotoxicity effect of essential oils on HaCaT.

Sample	LC_10_ (μL/mL)	LC_50_ (μL/mL)	LC_90_ (μL/mL)
Lavender F Bulgaria	0.70 ± 0.09	1.48 ± 0.16	2.26 ± 0.22
Lavender S Provence	0.27 ± 0.08	0.95 ± 0.29	1.63 ± 0.50
Mint F Spain	0.45 ± 0.15	2.02 ± 0.06	3.59 ± 0.03
Mint S India	0.40 ± 0.06	ND	ND

LC_10_—lethal concentration 10, LC_50_—lethal concentration 50, and LC_90_—lethal concentration 90. Lethal concentrations are expressed as volume percentages. ND—not detected in the tested range.

## Data Availability

Not applicable.

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
