# Peer review of "Antimicrobial Activity of Gelatin Nanofibers Enriched by Essential Oils against Cutibacterium acnes and Staphylococcus epidermidis"

_nanomaterials, 2023, doi:10.3390/nano13050844_

Round 1
Reviewer 1 Report
Renata Uhlířová et al. fabricated essential oils-loaded gelatin electrospun nanofiber patches, and mainly explored their anti-bacterial performances against C. acnes and S. epidermidis and biocompatibility to HaCaT cell line as potential antimicrobial patches for acne vulgaris local treatment. Some major revisions should be conducted before publication.
1. The Abstract section should be rewritten in a better and clear way. For instance, some important result data like antibacterial performance are suggested to be presented in a much more detailed manner.
2. Please state the reasons why gelatin was chosen in this study. What are the merits and advantages of gelatin compared with some other biopolymers like collagen, chitosan, etc.?
3. The introduction section was too long, which should be largely shortened in order to increase the readability.
4. The merits of electrospinning technique should be further outlined, and some recent works about the innovative electrospinning like European Polymer Journal, 2023, 186, 111863, https://doi.org/10.1016/j.eurpolymj.2023.111863 and ACS Applied Materials & Interfaces, 2022, 14(14), 15911-15926, https://doi.org/10.1021/acsami.1c24131 are suggested to be discussed.
5. Please state the reasons why 20% (w/w) EOs were added into the gelatin solution during electrospinning. Moreover, how did the authors choose the parameters of electrospinning? Do they conduct any preliminary experiments?
6. Why was 60% ethanol served as a negative control in MTT assay? Did the authors follow any standards?
7. The authors didn’t mention if the gelatin fibers were crosslinked or not. As known that, the gelatin component is water dissolvable. If the gelatin fibers were not crosslinked, the fiber morphology will be lost very quickly in the moisture environment. So how will the gelatin nanofibers be used as wound patches? How long will the patch maintain their morphology on the wound site?
8. The scale bars are not clear in the SEM images in Table 7, which should be redrawn.
9. How about the mechanical stability of as-prepared gelatin nanofibers? Did the addition of EOs affect the mechanical stability?
10. The grammar and writing should be improved in the whole manuscript. In addition, lot of reference errors and table errors should be double checked, and revised.
Author Response
Dear Reviewer,
thank you for your notes and questions.
Add 1 The abstract was rewritten in the suggested way.
Add 2 As we used electrospinning as the method for nanofiber fabrication collagen was excluded because, during the process of nanofiber preparation, it will degrade and turn into gelatin. Chitosan was used during some experiments, but the fabrication process was not as efficient as in the case of gelatin. Last but not least, poly-hydroxybutyrate was tested as well, but gelatin samples exceed it in almost all tested parameters (data not shown).
Add 3 The introduction section was shortened as suggested.
Add 4 Short discussion regarding some innovative electrospinning techniques was added into the Introduction session (ref. 30, 31). The list of references was updated.
Add 5 The addition of 20% of EOs was chosen because in the preliminary antimicrobial disc diffusion test 20% addition of EOs in almond oil exhibited adequate antimicrobial activity.
Add 6 The 60% ethanol was used as control according to our SOP standard operation protocol. Other alternatives will be DMSO, Triton X, or others.
Add 7 The gelatin fibers were not cross-linked. As mentioned in the abstract and in other parts of the manuscript, the goal was to prepare dissolvable patches. So, after contact with moisture on the skin, the nanofiber patch will dissolve and leave the EOs on the spot of application in adequate concentration. The patch will dissolve in the range of a few seconds.
Add 8 The SEM images were inserted again and the scale bars are now clearly visible.
Add 9 The addition of EOs did not alter mechanical stability. We were able to collect the fibers and manipulate them with no problem. Mechanical measurements were not performed.
Add 10 The manuscript was carefully double-checked and rewritten when it was appropriate.
I hope I have cleared up the inconsistencies and filled in the missing information.
Thank you once again.
Yours faithfully
Renata Uhlířová
Reviewer 2 Report
The paper discusses a novel, dissolvable patch to treat acne with essential oils. The paper is well thought out. The materials and methods all seem to be well described and in good order. The results, discussion and conclusion are all ok. The only comments I have are that the acronyms must be defined on first mention in the abstract and in the text. For example, MIC, MBC, MTT are not defined in the abstract; please check for such occurrences in the text as well. All figures and tables must stand on their own without reference to the text; make sure that all the acronyms are defined the figures and tables and that good descriptions are provided for each figure/table. English needs improvement. Although interesting, the introduction is probably too long.
Other comments:
Line 197 standart deviation…should be standard
Statistical analyses done?
Line 299 steered for should be stirred.
Electrospinning.
for 1 min with Au. What type of coater?
Line 321: Passaged – replated?
Line 329: placed back to the cultivator. Incubator. What is “pure medium”?
331 cultivator should be incubator.
Authors should read:
Solution blow spinning: A new method to produce micro‐and nanofibers from polymer solutions, E. S. Medeiros, G. M. Glenn, A. P. Klamczynski, W. J. Orts and L. H. Mattoso, Journal of applied polymer science 2009 Vol. 113 Issue 4 Pages 2322-2330
Author Response
Dear Reviewer,
thank you for your notes and questions. All errors and mistakes were corrected as suggested.
Statistical analysis is described in paragraph: 2.6 Statistical analysis.
The introduction was shortened. The text was checked carefully, the acronyms were defined in the abstract (MIC, MBC) and in the text as well. MTT assay is a cell viability assay using MTTT salt (1-(4,5-Dimethylthiazol-2-yl)-3,5-diphenylformazan, Thiazolyl blue formazan). More detailed descriptions were provided for each figure/table.
The type of used coater was Polaron SC7640 sputter coater.
The “pure medium” was only DMEM medium without any sample, maybe the proper description will be a fresh medium or just a DMEM medium as it included FBS as well as antibiotics and antimycotics.
Short discussion regarding some innovative electrospinning techniques was added into the Introduction session (ref. 32). The list of references was updated.
I hope I have cleared up the inconsistencies and filled in the missing information.
Thank you once again.
Yours faithfully
Renata Uhlířová
Round 2
Reviewer 1 Report
The reviewer's comments and concerns have been addressed.